# ADVERSARIAL SCORE MATCHING AND IMPROVED SAMPLING FOR IMAGE GENERATION

## ABSTRACT

Denoising Score Matching with Annealed Langevin Sampling (DSM-ALS) has recently found success in generative modeling. The approach works by first training a neural network to estimate the score of a distribution, and then using Langevin dynamics to sample from the data distribution assumed by the score network. Despite the convincing visual quality of samples, this method appears to perform worse than Generative Adversarial Networks (GANs) under the Fréchet Inception Distance, a standard metric for generative models. We show that this apparent gap vanishes when *denoising* the final Langevin samples using the score network. In addition, we propose two improvements to DSM-ALS: 1) Consistent Annealed Sampling as a more stable alternative to Annealed Langevin Sampling, and 2) a hybrid training formulation, composed of both Denoising Score Matching and adversarial objectives. By combining these two techniques and exploring different network architectures, we elevate score matching methods and obtain results competitive with state-of-the-art image generation on CIFAR-10.

## 1 INTRODUCTION

Song and Ermon (2019) recently proposed a novel method of generating samples from a target distribution through a combination of Denoising Score Matching (DSM) (Hyvärinen, 2005; Vincent, 2011; Raphan and Simoncelli, 2011) and Annealed Langevin Sampling (ALS) (Welling and Teh, 2011; Roberts et al., 1996). Since convergence to the distribution is guaranteed by the ALS, their approach (DSM-ALS) produces high-quality samples and guarantees high diversity. Though, this comes at the cost of requiring an iterative process during sampling, contrary to other generative methods. These generative methods can notably be used to diverse tasks like colorization, image restoration and image inpainting (Song and Ermon, 2019; Kadkhodaie and Simoncelli, 2020).

Song and Ermon (2020) further improved their approach by increasing the stability of score matching training and proposing theoretically sound choices of hyperparameters. They also scaled their approach to higher-resolution images and showed that DSM-ALS is competitive with other generative models. Song and Ermon (2020) observed that the images produced by their improved model were more visually appealing than the ones from their original work; however, the reported Fréchet Inception Distance (FID) (Heusel et al., 2017) did not correlate with this improvement.

Although DSM-ALS is gaining traction, Generative adversarial networks (GANs) (Goodfellow et al., 2014) remain the leading approach to generative modeling. GANs are a very popular class of generative models; they have been successfully applied to image generation (Brock et al., 2018; Karras et al., 2017; 2019; 2020) and have subsequently spawned a wealth of variants (Radford et al., 2015a; Miyato et al., 2018; Jolicoeur-Martineau, 2018; Zhang et al., 2019). The idea behind this method is to train a Discriminator ($D$) to correctly distinguish real samples from fake samples generated by a second agent, known as the Generator ($G$). GANs excel at generating high-quality samples as the discriminator captures features that make an image plausible, while the generator learns to emulate them.

Still, GANs often have trouble producing data from all possible modes, which limits the diversity of the generated samples. A wide variety of tricks have been developed to address this issue in GANs (Kodali et al., 2017; Gulrajani et al., 2017; Arjovsky et al., 2017; Miyato et al., 2018; Jolicoeur-Martineau and Mitliagkas, 2019), though it remains an issue to this day. DSM-ALS, on the other hand, does not suffer from that problem since ALS allows for sampling from the full distribution

captured by the score network. Nevertheless, the perceptual quality of DSM-ALS higher-resolution images has so far been inferior to that of GAN-generated images. Generative modeling has since seen some incredible work from Ho et al. (2020), who achieved exceptionally low (better) FID on image generation tasks. Their approach showcased a diffusion-based method (Sohl-Dickstein et al., 2015; Goyal et al., 2017) that shares close ties with DSM-ALS, and additionally proposed a convincing network architecture derived from Salimans et al. (2017).

In this paper, after introducing the necessary technical background in the next section, we build upon the work of Song and Ermon (2020) and propose improvements based on theoretical analyses both at training and sampling time. Our contributions are as follows:

- We propose Consistent Annealed Sampling (CAS) as a more stable alternative to ALS, correcting inconsistencies relating to the scaling of the added noise;
- We show how to recover the *expected denoised sample* (EDS) and demonstrate its unequivocal benefits *w.r.t* the FID. Notably, we show how to resolve the mismatch observed in DSM-ALS between the visual quality of generated images and its high (worse) FID;
- We propose to further exploit the EDS through a hybrid objective function, combining GAN and Denoising Score Matching objectives, thereby encouraging the EDS of the score network to be as realistic as possible.

In addition, we show that the network architecture used used by Ho et al. (2020) significantly improves sample quality over the RefineNet (Lin et al., 2017a) architecture used by Song and Ermon (2020). In an ablation study performed on CIFAR-10 and LSUN-church, we demonstrate how these contributions bring DSM-ALS in range of the state-of-the-art for image generation tasks *w.r.t.* the FID. The code to replicate our experiments is publicly available at **[Available in supplementary material]**.

## 2 BACKGROUND

### 2.1 DENOISING SCORE MATCHING

Denoising Score Matching (DSM) (Hyvärinen, 2005) consists of training a score network to approximate the gradient of the log density of a certain distribution ($\nabla_{\boldsymbol{x}} \log p(\boldsymbol{x})$), referred to as the score function. This is achieved by training the network to approximate a noisy surrogate of $p$ at multiple levels of Gaussian noise corruption (Vincent, 2011). The score network $s$, parametrized by $\theta$ and conditioned on the noise level $\sigma$, is tasked to minimize the following loss:

$$\frac{1}{2} \mathbb{E}_{p(\tilde{\boldsymbol{x}}, \boldsymbol{x}, \sigma)} \left[ \left\| \sigma s_\theta(\tilde{\boldsymbol{x}}, \sigma) + \frac{\tilde{\boldsymbol{x}} - \boldsymbol{x}}{\sigma} \right\|_2^2 \right], \tag{1}$$

where $p(\tilde{\boldsymbol{x}}, \boldsymbol{x}, \sigma) = q_\sigma(\tilde{\boldsymbol{x}}|\boldsymbol{x})p(\boldsymbol{x})p(\sigma)$. We define further $q_\sigma(\tilde{\boldsymbol{x}}|\boldsymbol{x}) = \mathcal{N}(\tilde{\boldsymbol{x}}|\boldsymbol{x}, \sigma^2 \boldsymbol{I})$ the corrupted data distribution, $p(\boldsymbol{x})$ the training data distribution, and $p(\sigma)$ the uniform distribution over a set $\{\sigma_i\}$ corresponding to different levels of noise. In practice, this set is defined as a geometric progression between $\sigma_1$ and $\sigma_L$ (with $L$ chosen according to some computational budget):

$$\{\sigma_i\}_{i=1}^L = \left\{ \gamma^i \sigma_1 \;\middle|\; i \in \{0, \ldots, L-1\}, \gamma \triangleq \frac{\sigma_2}{\sigma_1} = \ldots = \left(\frac{\sigma_L}{\sigma_1}\right)^{\frac{1}{L-1}} < 1 \right\}. \tag{2}$$

Rather than having to learn a different score function for every $\sigma_i$, one can train an unconditional score network by defining $s_\theta(\tilde{\boldsymbol{x}}, \sigma_i) = s_\theta(\tilde{\boldsymbol{x}})/\sigma_i$, and then minimizing Eq. 1. While unconditional networks are less heavy computationally, it remains an open question whether conditioning helps performance. Li et al. (2019) and Song and Ermon (2020) found that the unconditional network produced better samples, while Ho et al. (2020) obtained better results than both of them using a conditional network. Additionally, the denoising autoencoder described in Lim et al. (2020) gives evidence supporting the benefits of conditioning when the noise becomes small (also see App. D and E for a theoretical discussion of the difference). While our experiments are conducted with unconditional networks, we believe our techniques can be straightforwardly applied to conditional networks; we leave that extension for future work.

## 2.2 Annealed Langevin Sampling

Given a score function, one can use Langevin dynamics (or Langevin sampling) (Welling and Teh, 2011) to sample from the corresponding probability distribution. In practice, the score function is generally unknown and estimated through a score network trained to minimize Eq. 1. Song and Ermon (2019) showed that Langevin sampling has trouble exploring the full support of the distribution when the modes are too far apart and proposed Annealed Langevin Sampling (ALS) as a solution. ALS starts sampling with a large noise level and progressively anneals it down to a value close to 0, ensuring both proper mode coverage and convergence to the data distribution. Its precise description is shown in Algorithm 1.

---

**Algorithm 1** Annealed Langevin Sampling

**Require:** $s_\theta, \{\sigma_i\}_{i=1}^L, \epsilon, n_\sigma$.
1: Initialize $\boldsymbol{x}$

2: **for** $i \leftarrow 1$ to $L$ **do**
3: $\quad \alpha_i \leftarrow \epsilon \, \sigma_i^2 / \sigma_L^2$
4: $\quad$ **for** $n_\sigma$ **steps do**
5: $\quad\quad$ Draw $\boldsymbol{z} \sim \mathcal{N}(0, \boldsymbol{I})$
6: $\quad\quad \boldsymbol{x} \leftarrow \boldsymbol{x} + \alpha_i s_\theta(\boldsymbol{x}, \sigma_i) + \sqrt{2\alpha_i}\boldsymbol{z}$
$\quad$ **return** $\boldsymbol{x}$

**Algorithm 2** Consistent Annealed Sampling

**Require:** $s_\theta, \{\sigma_i\}_{i=1}^L, \gamma, \epsilon, \sigma_{L+1} = 0$
1: Initialize $\boldsymbol{x}$

2: $\beta \leftarrow \sqrt{1 - \left(1 - \epsilon/\sigma_L^2\right)^2 / \gamma^2}$
3: **for** $i \leftarrow 1$ to $L$ **do**
4: $\quad \alpha_i \leftarrow \epsilon \, \sigma_i^2 / \sigma_L^2$
5: $\quad$ Draw $\boldsymbol{z} \sim \mathcal{N}(0, \boldsymbol{I})$
6: $\quad \boldsymbol{x} \leftarrow \boldsymbol{x} + \alpha_i s_\theta(\boldsymbol{x}, \sigma_i) + \beta \sigma_{i+1}\boldsymbol{z}$
$\quad$ **return** $\boldsymbol{x}$

---

## 2.3 Expected denoised sample (EDS)

A little known fact from Bayesian literature is that one can recover a denoised sample from the score function using the Empirical Bayes mean (Robbins, 1955; Miyasawa, 1961; Raphan and Simoncelli, 2011):

$$s^*(\tilde{\boldsymbol{x}}, \sigma) = \frac{H^*(\tilde{\boldsymbol{x}}, \sigma) - \tilde{\boldsymbol{x}}}{\sigma^2}, \tag{3}$$

where $H^*(\tilde{\boldsymbol{x}}, \sigma) \triangleq \mathbb{E}_{\boldsymbol{x} \sim q_\sigma(\boldsymbol{x}|\tilde{\boldsymbol{x}})}[\boldsymbol{x}]$ is the expected denoised sample given a noisy sample (or Empirical Bayes mean), conditioned on the noise level. A different way of reaching the same result is through the closed-form of the optimal score function, as presented in Appendix D. The corresponding result for unconditional score function is presented in Appendix E for completeness.

The EDS corresponds to the expected real image given a corrupted image; it can be thought of as what the score network believes to be the true image concealed within the noisy input. It has also been suggested that denoising the samples (i.e., taking the EDS) at the end of the Langevin sampling improves their quality (Saremi and Hyvarinen, 2019; Li et al., 2019; Kadkhodaie and Simoncelli, 2020). In Section 4, we provide further evidence that denoising the final Langevin sample brings it closer to the assumed data manifold. In particular, we show that the Fréchet Inception Distance (FID) consistently decreases (improves) after denoising. Finally, in Section 5, we build a hybrid training objective using the properties of the EDS discussed above.

There are interesting links to be made between ALS and the RED algorithm (Romano et al., 2017; Reehorst and Schniter, 2018). The RED algorithm attempts to find the maximum a posteriori probability (MAP) denoised sample (i.e., the most plausible real data) given a noisy sample. It does so by solving an optimization problem to obtain a sample close to the noisy sample for which the EDS is a fixed point (denoising the sample does not change it because it is a real sample). Thus, just like ALS, the RED algorithm generates plausible real data given a score network. However, this algorithm does not ensure that we sample from the distribution and obtain full mode coverage. Thus, ALS's key benefit is ensuring that we sample from the full support of the distribution.

## 3 Consistent scaling of the noise

In this section, we present inconsistencies in ALS relating to the noise scaling and introduce Consistent Annealed Sampling (CAS) as an alternative.

## 3.1 INCONSISTENCIES IN ALS

One can think of the ALS algorithm as a sequential series of Langevin Dynamics (inner loop in Algorithm 1) for decreasing levels of noise (outer loop). If allowed an infinite number of steps $n_\sigma$, the sampling process will properly produce samples from the data distribution.

In ALS, the score network is conditioned on geometrically decreasing noise ($\sigma_i$). In the unconditional case, this corresponds to dividing the score network by the noise level (*i.e.*, $s_\theta(\tilde{\boldsymbol{x}}, \sigma_i) = s_\theta(\tilde{\boldsymbol{x}})/\sigma_i$). Thus, in both conditional and unconditional cases, we make the assumption that the noise of the sample at step $i$ will be of variance $\sigma_i^2$, an assumption upon which the quality of the estimation of the score depends. While choosing a geometric progression of noise levels seems like a reasonable (though arbitrary) schedule to follow, we show that ALS does not ensure such schedule.

Assume we have the true score function $s^*$ and begin sampling using a real image with some added zero-centered Gaussian noise of standard deviation $\sigma_0 = 50$. In Figure 1a, we illustrate how the intensity of the noise in the sample evolves through ALS and CAS, our proposed sampling, for a given sampling step size $\epsilon$ and a geometric schedule in this idealized scenario. We note that, although a large $n_\sigma$ approaches the real geometric curve, it will only reach it at the limit ($n_\sigma \to \infty$ and $\epsilon \to 0$). Most importantly, Figure 1b highlights how even when the annealing process does converge, the progression of the noise is never truly geometric; we prove this formally in Proposition 1.

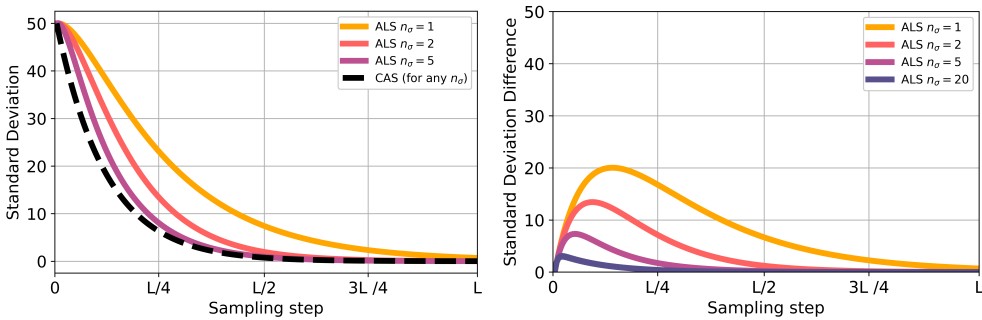

(a) Standard deviation of the noise in the image  (b) Difference between the standard deviation from ALS and CAS

Figure 1: Standard deviation during idealized sampling using a perfect score function $s^*$. The black curve in (a) corresponds to the true geometric progression, as demonstrated in Proposition 2.

**Proposition 1.** *Let $s^*$ be the optimal score function from Eq. 3. Following the sampling described in Algorithm 1, the variance of the noise component in the sample $\boldsymbol{x}$ will remain greater than $\sigma_t^2$ at every step $t$.*

The proof is presented in Appendix F. In particular, for $n_\sigma < \infty$, sampling has not fully converged and the remaining noise is carried over to the next iteration of Langevin Sampling. It also follows that for any $s_\theta$ different from the optimal $s^*$, the actual noise at every iteration is expected to be even higher than for the best possible score function $s^*$.

## 3.2 ALGORITHM

We propose Consistent Annealed Sampling (CAS) as a sampling method that ensures the noise level will follow a prescribed schedule for any sampling step size $\epsilon$ and number of steps $L$. Algorithm 2 illustrates the process for a geometric schedule. Note that for a different schedule, $\beta$ will instead depend on the step $t$, as in the general case, $\gamma_t$ is defined as $\sigma_{t+1}/\sigma_t$.

**Proposition 2.** *Let $s^*$ be the optimal score function from Eq. 3. Following the sampling described in Algorithm 2, the variance of the noise component in the sample $\boldsymbol{x}$ will consistently be equal to $\sigma_t^2$ at every step $t$.*

The proof is presented in Appendix G. Importantly, Proposition 2 holds no matter how many steps $L$ we take to decrease the noise geometrically. For ALS, $n_\sigma$ corresponds to the number of steps

per level of noise. It plays a similar role in CAS: we simply dilate the geometric series of noise levels used during training by a factor of $n_\sigma$, such that $L_{\text{sampling}} = (L_{\text{training}} - 1)n_\sigma + 1$. Note that the proposition only holds when the initial sample is a corrupted image (*i.e.*, $\boldsymbol{x}_0 = \mathcal{I} + \sigma_0 \boldsymbol{z}_0$). However, by defining $\sigma_0$ as the maximum Euclidean distance between all pairs of training data points (Song and Ermon, 2020), the noise becomes in practice much greater than the true image; sampling with pure noise initialization (*i.e.*, $\boldsymbol{x}_0 = \sigma_0 \boldsymbol{z}_t$) becomes indistinguishable from sampling with data initialization.

## 4 BENEFITS OF THE EDS ON SYNTHETIC DATA AND IMAGE GENERATION

As previously mentioned, it has been suggested that one can obtain better samples (closer to the assumed data manifold) by taking the EDS of the last Langevin sample. We provide further evidence of this with synthetic data and standard image datasets.

It can first be observed that the sampling steps correspond to an interpolation between the previous point and the EDS, followed by the addition of noise.

**Proposition 3.** *Given a noise-conditional score function, the update rules from Algorithm 1 and Algorithm 2 are respectively equivalent to the following update rules:*

$$\boldsymbol{x} \leftarrow (1 - \eta)\boldsymbol{x} + \eta H(\boldsymbol{x}, \sigma_i) + \sqrt{2\eta}\sigma_i \boldsymbol{z} \qquad \text{for } \boldsymbol{z} \sim \mathcal{N}(0, \boldsymbol{I}) \text{ and } \eta = \frac{\epsilon}{\sigma_L^2}$$

$$\boldsymbol{x} \leftarrow (1 - \eta)\boldsymbol{x} + \eta H(\boldsymbol{x}, \sigma_i) + \beta \sigma_{i+1} \boldsymbol{z}$$

The demonstration is in Appendix H. This result is equally true for an unconditional score network, with the distinction that $\eta$ would no longer be independent of $\sigma_i$ but rather linearly proportional to it.

Intuitively, this implies that the sampling steps slowly move the current sample towards a moving target (the EDS). If the sampling behaves appropriately, we expect the final sample $\boldsymbol{x}$ to be very close to the EDS, *i.e.*, $\boldsymbol{x} \approx H(\boldsymbol{x}, \sigma_L)$. However, if the sampling step size is inappropriate, or if the EDS does not stabilize to a fixed point near the end of the sampling, these two quantities may be arbitrarily far from one another. As we will show, the FIDs from Song and Ermon (2020) suffer from such distance.

From Proposition 3, we see that CAS shares some similarities with the algorithm by Kadkhodaie and Simoncelli (2020). While the weight we give to the denoiser ($\eta$) decreases geometrically (by its linearity in $\sigma$), their schedule appears to be much steeper. They also estimate the residual noise in their samples by the $l_2$ norm instead of determining it through a schedule, as CAS strives to do. As a note, we had found weak evidence during development that estimating the residual noise worsened the FID.

The equivalence showed in Proposition 3 suggests instead to take the expected denoised sample at the end of the Langevin sampling as the final sample; this would be equivalent to the update rule $\boldsymbol{x} \leftarrow H(\boldsymbol{x}, \sigma_L)$ at the last step. Synthetic 2D examples shown in Figure 2 demonstrate the immediate benefits of this technique.

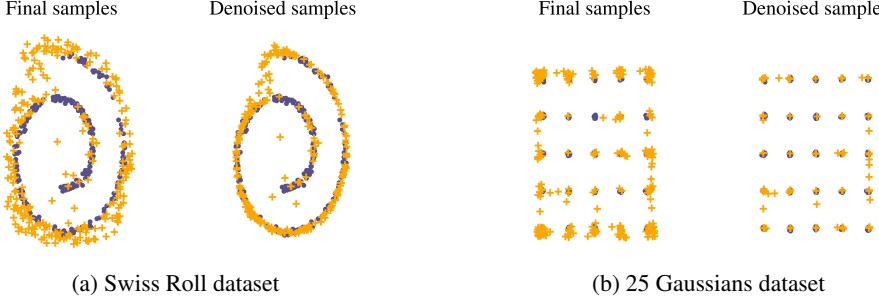

(a) Swiss Roll dataset          (b) 25 Gaussians dataset

Figure 2: Langevin sampling on synthetic 2D experiments. Circles are real data points, crosses are generated data points. On both datasets, taking the EDS brings the samples much closer to the real data manifold.

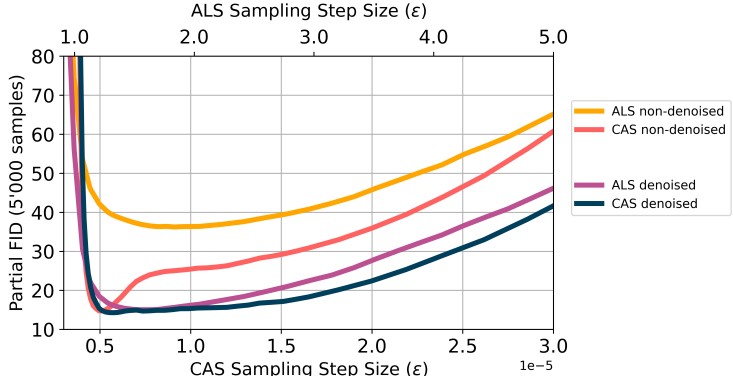

Figure 3: Partial estimate of FID (lower is better) as a function of the sampling step size on CIFAR-10, with $n_\sigma = 1$. The interactions between *consistent sampling* and denoising are shown.

We train a score network on CIFAR-10 (Krizhevsky et al., 2009) and report the FID from both ALS and CAS as a function of the sampling step size and of denoising in Figure 3. The first observation to be made is just how critical denoising is to the FID score for ALS, even as its effect cannot be perceived by the human eye. For CAS, we note that the score remains small for a much wider range of sampling step sizes when denoising. Alternatively, the sampling step size must be very carefully tuned to obtain results close to the optimal.

Figure 3 also shows that, with CAS, the FID of the final sample is approximately equal to the FID of the denoised samples for small sampling step sizes. Furthermore, we see a smaller gap in FID between denoised and non-denoised for larger sampling step sizes than ALS. This suggests that consistent sampling is resulting in the final sample being closer to the assumed data manifold (i.e., $\boldsymbol{x} \approx H_\theta(\boldsymbol{x}, \sigma_L)$).

Interestingly, when Song and Ermon (2020) improved their score matching method, they could not explain why the FID of their new model did not improve even though the generated images looked better visually. To resolve that matter, they proposed the use of a new metric (Zhou et al., 2019) that did not have this issue. As shown in Figure 3, denoising resolves this mismatch.

## 5 ADVERSARIAL FORMULATION

The score network is trained to recover an uncorrupted image from a noisy input minimizing the $l_2$ distance between the two. However, it is well known from the image restoration literature that $l_2$ does not correlate well with human perception of image quality (Zhang et al., 2012; Zhao et al., 2016). One way to take advantage of the EDS would be to encourage the score network to produce an EDS that is more realistic from the perspective of a discriminator. Intuitively, this would incentivize the score network to produce more discernible features at inference time.

We propose to do so by training the score network to simultaneously minimize the score-matching loss function and maximize the probability of denoised samples being perceived as real by a discriminator. We use alternating gradient descent to sequentially train a discriminator for a determined number of steps at every score function update.

In our experiments, we selected the Least Squares GAN (LSGAN) (Mao et al., 2017) formulation as it performed best (see Appendix B for details). For an unconditional score network, the objective functions are as follows:

$$\max_\phi \ \mathbb{E}_{p(\boldsymbol{x})} \left[ (D_\phi(\boldsymbol{x}) - 1)^2 \right] + \mathbb{E}_{p(\tilde{\boldsymbol{x}}, \boldsymbol{x}, \sigma)} \left[ (D_\phi(H_\theta(\tilde{\boldsymbol{x}}, \sigma) + 1)^2 \right] \tag{4}$$

$$\min_\theta \ \mathbb{E}_{p(\tilde{\boldsymbol{x}}, \boldsymbol{x}, \sigma)} \left[ (D_\phi(H_\theta(\tilde{\boldsymbol{x}}, \sigma)) - 1)^2 + \frac{\lambda}{2} \left\| \sigma s_\theta(\tilde{\boldsymbol{x}}, \sigma) + \frac{\tilde{\boldsymbol{x}} - \boldsymbol{x}}{\sigma} \right\|_2^2 \right], \tag{5}$$

where $H_\theta(\tilde{\boldsymbol{x}}, \sigma) = s_\theta(\tilde{\boldsymbol{x}}, \sigma)\sigma^2 + \tilde{\boldsymbol{x}}$ is the EDS derived from the score network. Eq. 4 is the objective function of the LSGAN discriminator, while Eq. 5 is the adversarial objective function of the score network derived from Eq. 1 and from the LSGAN objective function.

We note the similarities between these objective functions and those of an LSGAN adversarial autoencoder (Makhzani et al., 2015; Tolstikhin et al., 2017; Tran et al., 2018), with the distinction of using a denoising autoencoder $H$ as opposed to a standard autoencoder. We can highlight this difference by reformulating Eq. 5 as:

$$\min_\theta \ \mathbb{E}_{p(\tilde{\boldsymbol{x}}, \boldsymbol{x}, \sigma)} \left[ (D_\phi(H_\theta(\tilde{\boldsymbol{x}}, \sigma)) - 1)^2 + \frac{\lambda}{2\sigma^2} \|H_\theta(\tilde{\boldsymbol{x}}, \sigma) - \boldsymbol{x}\|_2^2 \right], \tag{6}$$

As GANs favor quality over diversity, there is a concern that this hybrid objective function might decrease the diversity of samples produced by the ALS. In Section 6.1, we first study image generation improvements brought by this method and then address the diversity concerns with experiments on the 3-StackedMNIST (Metz et al., 2016) dataset in Section 6.2.

# 6 EXPERIMENTS

## 6.1 ABLATION STUDY

We ran experiments on CIFAR-10 (Krizhevsky et al., 2009) and LSUN-churches (Yu et al., 2015) with the score network architecture used by Song and Ermon (2020). We also ran similar experiments with an unconditional version of the network architecture by Ho et al. (2020), given that their approach is similar to Song and Ermon (2019) and they obtain very small FIDs. For the hybrid adversarial score matching approach, we used an unconditional BigGAN discriminator (Brock et al., 2018). We compared three factors in an ablation study: adversarial training, Consistent Annealed Sampling and denoising.

Details on how the experiments were conducted are found in Appendix B. Unsuccessful experiments with large images are also discussed in Appendix C. See also Appendix I for a discussion pertaining to the use of the Inception Score (Heusel et al., 2017), a popular metric for generative models.

Results for CIFAR-10 and LSUN-churches with Song and Ermon (2019) score network architecture are respectively shown in Table 1 and 2. Results for CIFAR-10 with Ho et al. (2020) score network architecture are shown in Table 3.

Table 1: [Non-denoised / Denoised FID] from 10k samples on CIFAR-10 (32x32) with Song and Ermon (2019) score network architecture

| Sampling | Non-adversarial | Adversarial |
|---|---|---|
| non-consistent ($n_\sigma = 1$) | 36.3 / 13.3 | 30.0 / 11.8 |
| non-consistent ($n_\sigma = 5$) | 33.7 / 10.9 | 26.4 / **9.5** |
| consistent ($n_\sigma = 1$) | 14.7 / 12.3 | 11.9 / 10.8 |
| consistent ($n_\sigma = 5$) | 12.7 / 11.2 | 9.9 / 9.7 |

Table 2: [Non-denoised / Denoised FID] from 10k samples on LSUN-Churches (64x64) with Song and Ermon (2019) score network architecture

| Sampling | Non-adversarial | Adversarial |
|---|---|---|
| non-consistent ($n_\sigma = 1$) | 43.2 / 40.3 | 40.9 / 36.7 |
| non-consistent ($n_\sigma = 5$) | 42.0 / 39.2 | 40.0 / 35.8 |
| consistent ($n_\sigma = 1$) | 41.5 / 40.7 | 38.2 / 36.7 |
| consistent ($n_\sigma = 5$) | 39.5 / 39.1 | 36.3 / **35.4** |

We always observe an improvement in FID from denoising and by increasing $n_\sigma$ from 1 to 5. We observe an improvement from using the adversarial approach with Song and Ermon (2019) network

Table 3: [Non-denoised / Denoised FID] from 10k samples on CIFAR-10 (32x32) with Ho et al. (2020) unconditional score network architecture

| Sampling | Non-adversarial | Adversarial |
|---|---|---|
| non-consistent ($n_\sigma = 1$) | 25.3 / 7.5 | 21.6 / 7.5 |
| non-consistent ($n_\sigma = 5$) | 20.0 / **5.6** | 17.7 / 6.1 |
| consistent ($n_\sigma = 1$) | 7.8 / 7.1 | 7.7 / 7.1 |
| consistent ($n_\sigma = 5$) | 6.2 / 6.1 | 6.1 / 6.5 |

architecture, but not on denoised samples with the Ho et al. (2020) network architecture. We hypothesize that this is a limitation of the architecture of the discriminator since, as far as we know, no variant of BigGAN achieves an FID smaller than 6. Nevertheless, it remains advantageous for more simple architectures, as shown in Table 1 and 2. We observe that consistent sampling outperforms non-consistent sampling on the CIFAR-10 task at $n_\sigma = 1$, the quickest way to sample.

We calculated the FID of the non-consistent denoised models from 50k samples in order to compare our method with the recent work from Ho et al. (2020). We obtained a score of 3.65 for the non-adversarial method and 4.02 for the adversarial method on the CIFAR-10 task when sharing their architecture; these scores are close to their reported 3.17. Although not explicit in their approach, Ho et al. (2020) denoised their final sample. This suggests that taking the EDS and using an architecture akin to theirs were the two main reasons for outperforming Song and Ermon (2020). Of note, our method only trains the score network for 300k iterations, while Ho et al. (2020) trained their networks for more than 1 million iterations to achieve similar results.

## 6.2 NON-ADVERSARIAL AND ADVERSARIAL SCORE NETWORKS HAVE EQUALLY HIGH DIVERSITY

To assess the diversity of generated samples, we evaluate our models on the 3-Stacked MNIST generation task (Metz et al., 2016) (128k images of 28x28), consisting of numbers from the MNIST dataset (LeCun et al., 1998) superimposed on 3 different channels. We trained non-adversarial and adversarial score networks in the same way as the other models. The results are shown in Table 4.

We see that each of the 1000 modes is covered, though the KL divergence is still inferior to PACGAN (Lin et al., 2018), meaning that the mode proportions are not perfectly uniform. Blindness to mode proportions is thought to be a fundamental limitation of score-based methods (Wenliang, 2020). Nevertheless, these results confirm a full mode coverage on a task where most GANs struggle and, most importantly, that using a hybrid objective does not hurt the diversity of the generated samples.

| 3-Stacked MNIST | | |
|---|---|---|
| | Modes (Max 1000) | KL |
| DCGAN (Radford et al., 2015b) | 99.0 | 3.40 |
| ALI (Dumoulin et al., 2016) | 16.0 | 5.40 |
| Unrolled GAN (Metz et al., 2016) | 48.7 | 4.32 |
| VEEGAN (Srivastava et al., 2017) | 150.0 | 2.95 |
| PacDCGAN2 (Lin et al., 2017b) | 1000.0 | 0.06 |
| WGAN-GP (Kumar et al., 2019; Gulrajani et al., 2017) | 959.0 | 0.73 |
| PresGAN (Dieng et al., 2019) | 999.6 | 0.115 |
| MEG (Kumar et al., 2019) | 1000.0 | 0.03 |
| Non-adversarial DSM (ours) | 1000.0 | 1.36 |
| Adversarial DSM (ours) | 1000.0 | 1.49 |

Table 4: As in Lin et al. (2018), we generated 26k samples and evaluated the mode coverage and KL divergence based on the predicted modes from a pre-trained MNIST classifier.

## 7   CONCLUSION

We proposed Consistent Annealed Sampling as an alternative to Annealed Langevin Sampling, which ensures the expected geometric progression of the noise and brings the final samples closer to the data manifold. We showed how to extract the expected denoised sample and how to use it to further improve the final Langevin samples. We proposed a hybrid approach between GAN and score matching. With experiments on synthetic and standard image datasets; we showed that these approaches generally improved the quality/diversity of the generated samples.

We found equal diversity (coverage of all 1000 modes) for the adversarial and non-adversarial variant of the difficult StackedMNIST problem. Since we also observed better performance (from lower FIDs) in our other adversarial models trained on images, we conclude that making score matching adversarial increases the quality of the samples without decreasing diversity. These findings imply that score matching performs better than most GANs and on-par with state-of-the-art GANs. Furthermore, our results suggest that hybrid methods, combining multiple generative techniques together, are a very promising direction to pursue.

As future work, these models should be scaled to larger batch sizes on high-resolution images, since GANs have been shown to produce outstanding high-resolution images at very large batch sizes (2048 or more). We also plan to further study the theoretical properties of CAS by considering its corresponding stochastic differential equation.

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
