# OpenReview forum: "Adversarial score matching and improved sampling for image generation"
_ICLR.cc/2021/Conference — ICLR 2021 Poster_

### Official Review · AnonReviewer2 · 2020-10-27
**Review round 0 for “ Adversarial score matching and improved sampling for image generation”**

**Rating:** 7
**Confidence:** 2

**Review:**

The article deals with generative models based on “Annealed Langevin Sampling“ rather than a GAN.
Theses models suffer from worse FID than GAN.
Authors proposed to denoise the last Langevin samples to reduce the gap in performance with Adversarial Network.
The paper is really easy to read with good illustrations and supporting experiments.

In order to gain in comprehension, especially for people new to ALS, it would have been great if authors have proposed an illustration (and comparison) of the samples evolution along Alg 1 and Alg 2 .

Authors are honest in their revised results comments but I don’t known if they will be able to include the erratum  in a final version

As I was not aware before this review of “Annealed Langevin Sampling” my rating may not be confident.

---

> ### Author Response · Authors · 2020-11-24
> **AnonReviewer2**
>
> Thank you for your comments!
>
> As per your suggestion, we added Figure 4 (p18) to show the samples evolving over time for the consistent and non-consistent sampling in the non-adversarial and adversarial setting.

---

### Official Review · AnonReviewer4 · 2020-10-28
**Recommendation to Accept**

**Rating:** 7
**Confidence:** 3

**Review:**

The paper presents a novel approach for denoising score matching, where the Annealed Langevin Sampling has been substituted by Consistent Annealed Sampling, which adds more stability to the process.

The paper is in general clear and well-written. The contributions are clearly highlighted and the proposed approach is conveniently compared with other state of the art methods, demonstrating its superiority.

Positive aspects:
- The Consistent Annealed Sampling proposed in this paper is more stable than the Annealed Langevin Sampling
- The combination between GAN and score matching improves the quality/diversity of the generated sample

Negative aspects:
- The limitation of the method to Gaussian noise
- The presentation of a real scenario for your approach would have been a plus

However, I have some questions:
1. Who is n_sigma parameter in Algorithm 1?
2. Algorithm 1, line 4: there is no iteration over 't' in the loop?
3. How does your denoising scheme work? Do you create noisy samples from your real data and try to denoise them using the proposed approach? Because taking a sample affected by random noise (in the test phase) I guess it won't work.
4. The denoising scheme is used in a GAN framework, the denoised samples being perceived as real by the discriminator. Is the system trained end-to-end or first you denoise the image and afterwards you train the GAN?
5. Could you please indicate an application scenario which could benefit from this approach, e.g. image-to-image translation, domain adaptation, etc.?
6. Your method is assuming Gaussian noise. Can it be extended to the case of general noise (a noise model which could be also learnt)?

---

> ### Author Response · Authors · 2020-11-24
> **AnonReviewer4**
>
> Thank you for your questions! Let us begin by addressing the negative aspects:
>
> Negative aspects:
> - Gaussian noise is essential for the Langevin process to function. Though, it would be possible to imagine a different diffusion process coupled with a different MC sampler. However, note that the literature on continuous stochastic processes in the space of real numbers that do not rely on the Brownian motion (leading to Gaussian noise) is scarce. We are not aware of a stochastic process that results in provable sampling that does not rely on the Brownian motion.
> - Mentioning real scenario usages for generative methods would indeed help justify their importance. We have added a sentence in the first paragraph of the introduction to reflect this.
>
> Answers:
>
> 1) $n_\sigma$ is either set to 1 or 5. It is specified in the experiment section.
> 2) That’s right. We’ve changed the corresponding line to “for $n_\sigma$ steps do: “ to reflect this better.
> 3) During the training, the denoiser is only used to train the discriminator. Here, the “fake” samples are produced by taking a real sample from the training dataset, corrupting it, and then using the score network for the denoising scheme described in Section 4. The discriminator is then tasked to distinguish between these reconstructions and the original images.
> 4) The discriminator and the generator (here, the score network) are trained concurrently. The second paragraph of Section 5 mentions that the discriminator is trained for $n_D$ steps for every score network step. $n_D$ is specified in the third paragraph of Appendix B.
> 5) See second bullet point.
> 6) See first bullet point.

---

### Official Review · AnonReviewer1 · 2020-10-28
**Interesting finding about ALS, other contributions are less convincing**

**Rating:** 6
**Confidence:** 3

**Review:**

The submission presents three contributions. First, the authors show the inconsistencies in the existing annealed Langevin sampling used in score-matching generative models and propose to correct it with the newly proposed Consistent Annealed Sampling (CAS) algorithm. The second contribution claimed is in providing evidence of the benefits of Expected Denoised Sample (EDS). Furthermore, the submission introduces a hybrid adversarial score-matching model that demonstrates improvements in terms of FID on simpler architectures.

The proposed CAS algorithm is theoretically well-motivated based on the observation that ALS is inconsistent with the scaling of the noise during sampling process (although the question whether noise should follow none other than geometric progression is still an open question). The paper is well-written, and the ablation study is carried out well.

However, it is a bit confusing as to whether the EDS (although under a different name — denoising jump) is a contribution of this paper or is something proposed prior to this work. I understand that this denoising procedure has already been presented as a necessary technique in score matching models. Nevertheless, I believe the authors contributed by showing that both ALS and CAS move samples towards the EDS (Proposition 3) and show additional empirical evidence of its benefits on synthetic and real datasets.

Taking EDS on the last Langevin step diminishes the impact of CAS (doesn't bring unambiguous improvement in FID scores in the experiments), otherwise very interesting finding both theoretically and algorithmically, and substitute for ALS.

The effect of the hybrid model is also not persistent and depends on the architecture used. For an incremental improvement (a combination of two models), the improvement is not consistent across architectures. The paper does not explain whether there is a good rationale for such a combination; therefore I remain sceptical about the results.

Given all the above, I am still leaning a bit towards accepting the paper as it covers an interesting finding relating to the ALS. Although the CAS effect on performance is limited by the EDS, score-matching models are of broad interest for the ICLR community.

---

> ### Author Response · Authors · 2020-11-24
> **AnonReviewer1**
>
> Thank you for your comments!
>
> We agree with your assessment of the EDS and our contributions to it.
>
> We were uncertain why the Ho et al. (2020) architecture did not see an improvement in denoised FID. However, at the time, we hypothesized that this may be a limitation of the discriminator as GANs never reach such low levels of FIDs (see p8). However, note that there was a bug affecting only the adversarial methods at the time of submission (we mentioned this in https://openreview.net/forum?id=eLfqMl3z3lq&noteId=f8yXT0iQCm). After fixing the bug, the denoised FID is now equal for the adversarial and non-adversarial methods except at “non-consistent (nσ = 5)” where the non-adversarial method still does a bit better (5.6 vs 6.1). In addition, the non-denoised FID is now always lower for the adversarial method (and the difference is very large for the non-consistent sampling). Thus, with Ho et al. (2020) architecture, the adversarial method still provides some benefits, rather than merely always performing worse (as it did before fixing the bug).
>
> The adversarial hybrid method is well justified considering the vast literature on adversarial autoencoders and autoencoders in feature space (rather than input space). We now highlight the connection between our hybrid approach and adversarial autoencoders more clearly on page 6.

---

### Official Review · AnonReviewer3 · 2020-11-02
**Improvement of Langevin dynamics to sample from score matching functions adversarially trained.**

**Rating:** 7
**Confidence:** 3

**Review:**

This paper tackles the problem of generative modeling by using Langevin dynamics to sample from the denoising score function. Recently, this family of approaches (Song and Ermon 2019, Song and Ermon 2020) has shown promising and competitive results being positioned as a potential alternative to GANs.

The paper introduces different improvements over Song and Ermon (2020). A different sampling dynamic (Consistent Annealed Sampling) that produces a more stable training that the traditional annealing scheme by carefully scaling the injected noise. Second, it is empirically shown that running a denoising step on the generated sample leads to an improvement of the FID score. Based on this observation, the paper proposes to use a denoiser trained in an adversarial fashion to synthesize more realistic images.

The work addresses the very relevant problem of how to synthesize images in a realistic way, introducing some modifications to existing works that lead to an improvement on the quality of the generated image.  The paper is well written, presents a nice introduction to the method, which allows to motivate the different modifications in a natural way. The proposed modifications are analyzed in low-dimensional toy experiments and in small-scale images (CIFAR, LSUN-churchers, Stacked-MNIST).

In what follows I list a few questions:

1. Would it be possible to analyze the strategy of sampling presented in Kadkhodaei and Simoncelli 2020 (concurrent work), and compare to the one proposed in the paper? Both strategies seem to improve the stabilization of the procedure by scaling the noise.

2. Regarding the step of applying the denoiser to the generated sample. I wonder what happens if the denoiser is re-applied? Also, is this connected to the fact that the denoiser may have a fixed point and this fixed point might lead to a better sample?

3. Regarding using an adversarial denoising. In the denoising literature, there are a few works connecting score matching and state-of-the-art image denoisers. I would like to see a better discussion of this. For example, see,

Romano, Y., Elad, M. and Milanfar, P., 2017. The little engine that could: Regularization by denoising (RED). SIAM Journal on Imaging Sciences, 10(4), pp.1804-1844.

Reehorst, E.T. and Schniter, P., 2018. Regularization by denoising: Clarifications and new interpretations. IEEE transactions on computational imaging, 5(1), pp.52-67.

---
After Discussion:
I think this is a good paper and I would like to see it presented at ICLR2021.

---

> ### Author Response · Authors · 2020-11-24
> **AnonReviewer3**
>
> Thank you for your questions! We address them below:
>
> 1) While their training methods differ a lot from ours, both their sampling method and our proposed CAS share some similarities. Namely, their Eq. 5 is comparable to the equation we show in Prop. 3 in the unconditional case, with key differences: while the weight we give to the denoiser (η) decreases geometrically (by its linearity in σ), their schedule appears to be much steeper. They also estimate the residual noise in their samples by the L2 norm instead of determining it through a schedule, as CAS strives to do. As a note, we had found weak evidence during development that estimating the residual noise worsened the FID. We now discuss the links between both algorithms on p5.
>
> 2) Empirically, applying the denoiser several times on images leads to a decrease in FID, while the differences are not perceptible to the human eye. This is because our denoiser is imperfectly conditioned on a non-zero noise variance. However, intuitively, if we had selected a wrong learning rate, applying the denoiser multiple times might help alleviate the FID penalty caused by the remaining imperceptible noise left at the end.
>
> 3) We found these papers very interesting given the links that can be made between them and our approach. As we seem to understand, the Red algorithm attempts to find the MAP denoised estimate (most plausible real data) and it makes use of the same denoiser that we have (based on Vincent denoising loss function). However, our goal is not to find the most plausible close real data as doing so would not ensure that we generate from all modes of the distribution. We use Langevin sampling in order to ensure that, rather than just obtaining the closest real data point, we move with enough randomness to sample from the full data distribution.  We now have a discussion on the RED algorithm on page 3.

---

### Author Response · Authors · 2020-11-24
**Addressing the reviewers**

We want to thank all the reviewers for their time and positive feedback. We have uploaded a new version of the paper taking into account their various suggestions (extra references, discussion of related work and Figure 4). We have also updated the numbers of the adversarial score matching in Tables 1, 2 and 3 (as per our comment https://openreview.net/forum?id=eLfqMl3z3lq&noteId=f8yXT0iQCm).

---

### Decision · Program_Chairs · 2021-01-07
**Final Decision**

**Decision:**

Accept (Poster)

**Comment:**

This paper introduces an alternative to Langevin sampling and also the idea of adversarial score sampling.
The reviewers are generally supportive of the paper.

Pros:
- The idea behind improving Langevin sampling is theoretically justified and leads to a simple algorithm.
- The idea behind adversarial score matching is also shown to be effective
- Improvement over baseline

Cons:
- Two ideas packed  into one paper, which is reflected by the title as well.
-  From the narrative it could be thought that using EDS on the last step of CAS is the contribution of the paper.